# Influence of Stirring Parameters on Creaminess of Spring Blossom Honey Measured by Crystal Size, Whiteness Index and Mouthfeel

**DOI:** 10.3390/foods12010048

**Published:** 2022-12-22

**Authors:** Mario Meixner, Mareike Weber, Sebastian Lella, Wilfried Rozhon, Margot Dasbach

**Affiliations:** Department Agriculture, Ecotrophology and Landscape Development of Anhalt University of Applied Sciences, Strenzfelder Allee 28, 06406 Bernburg, Germany

**Keywords:** honey, creaminess, honey stirring, crystal size, whiteness index, mouthfeel

## Abstract

Spring blossom honey from regions with many rape fields tends to crystalize rapidly after harvesting. The crystallization process needs to be controlled by stirring in order to avoid the formation of coarse crystals and to ensure the creaminess of honey. The aim of this study was to investigate how various parameters of the stirring process influence the creaminess of spring blossom honey in order to give recommendations for beekeeping practices. The creaminess was quantified by measuring the crystal size by microscopic analysis, measuring the whiteness index by color analysis using CIE Lab and by sensory analysis. We investigated the influence of five stirring parameters, including the type of stirring device, honey pretreatment, stirring temperature (14 °C to room temperature), stirring interval (1 to 24 times) and stirring time (1–15 min) on the creaminess of honey. We found that the stirring temperature is the most important factor for honey creaminess. At the optimal temperature of 14 °C, other factors like seed honey, stirring time and stirring interval have only a neglectable effect. If the optimal temperature of 14 °C cannot be maintained, as it may happen in beekeepers’ practice, sieving the honey with a mesh size of 200 µm before stirring, the addition of seed honey prepared with a kitchen food processor, and using a stirring screw and stirring several times per day is recommended.

## 1. Introduction

Honeybees forage nectar and honeydew from plants and ripe it to honey in the beehive. The beekeeper harvests the ripe honey as a liquid. Depending on the composition, liquid honey is more or less prone to crystallization. Honey consists predominantly of glucose and fructose [1], which sum up to approximately 70–80% (*w*/*w*), typically less than 20% (*w*/*w*) water and a number of other di- and trisaccharides in smaller quantities [2] as well as pollen, minerals and small amounts of phytochemicals. Based on the bee’s flowering forage, the ratio of the sugars varies [2,3,4], which has a major impact on the honey’s tendency to crystallize. Laos et al. [5], as well as Scripca & Amariei [6], found that the most important parameter for crystallization is the fructose/glucose ratio in favor of glucose. Scripca & Amariei [6] emphasize that also the glucose/water ratio is important for the crystallization behavior of honey. A saturated solution of fructose in pure water contains 780 g kg^−1^, while a saturated solution of glucose contains only 470 g kg^−1^ [7]. Thus, particularly glucose-rich honey is a supersaturated solution and crystallizes more rapidly and to a larger extent than fructose-rich honey. When crystallization has initiated, the honey gets opaque, and its viscosity increases significantly.

Most consumers prefer liquid or melt-in-the-mouth creamy honey with small crystals that are not noticeable on the tongue [5,6,8,9,10,11,12]. There are three possibilities to meet the customers’ preferences: delaying or avoiding crystallization, liquefying honey after crystallization or controlling the crystallization process by stirring. In order to avoid crystallization, honey is often filtered and pasteurized [11,13,14,15]. According to German food law, filtration of honey is not permitted [16]. Pasteurization reduces the activity of several enzymes, particularly invertase, and increases the HMF (5-hydroxymethylfurfural) content, which is undesirable. Recently, methods for preventing honey crystallization based on ultrasound [8,12,17,18] or freezing [19] have been described.

Crystallization of honey takes place in two phases: crystal nucleation and crystal growth [19,20]. The initial formation of crystallization nuclei happens on small crystals, pollen, dust grains or similar solids and air bubbles. Subsequently, glucose molecules accumulate, creating larger structures that network with one another (crystal growth). The more crystallization nuclei are present, the more individual and, therefore, finer crystals are formed during the crystallization process. The resulting creamy honey will have a melt-in-the-mouth feel. In contrast, if few crystallization nuclei are present, the sugar molecules adhere to one another in large numbers, which leads to the formation of large, coarse crystals noticeable on the tongue. To control the crystallization process, crystallization starters can be added, and honey can be stirred. However, the only crystallization starter that may be used is already crystallized honey since, according to German Food law, the addition of substances other than honey [21,22,23] is not permitted [16].

Stirring honey during crystallization is a usual practice of beekeepers to avoid coarse crystals and to ensure creamy and spreadable consistency. Stirring distributes the crystallization nuclei more uniformly and promotes the formation of small crystals. Pioneer Dyce [24] patented a process including short-time heating of honey up to 70 °C in order to melt all crystals, cooling to room temperature, adding 5% of creamy honey with small crystals, stirring the honey for 15 min and storing it at 14 °C for 34 h. Costa et al. [25] reported that the stirring temperature has a greater impact on crystal size than the stirring speed. The smallest crystals were found at a temperature of 15 °C and a stirring speed of 540 rpm. Radtke and Lichtenberg–Kraag [26] reported the homogenization of honey from the beginning of crystallization to a creamy consistency and the storage of honey in a cold room as optimal conditions. Tappi et al. [27] reported that controlling the crystallization process by adding crystallization nuclei in the form of well-crystallized seed honey and stirring leads to a uniform honey texture. However, the effect of the amount of seed honey and the stirring rate on the final honey quality remained elusive.

The aim of this study was to investigate how the factors of stirring device, stirring temperature, pretreatment of the honey before stirring, the addition of seed honey, stirring time and number of stirring intervals per day influence spring blossom honey creaminess. To simulate the conditions in beekeeping practice, we used 25 kg of honey for each experimental setup. This allowed the identification of the most suitable conditions for obtaining creamy honey and to give recommendations for the stirring process in beekeeping practice.

## 2. Materials and Methods

### 2.1. Material and Treatment of Honey

During the years 2019–2021, honey stirring tests were carried out with honey from the first harvest of the year in the teaching apiary of Anhalt university of applied sciences (Saxony-Anhalt, Germany). Due to many rape fields in the area blooming before the first harvest, the honey is rich in glucose, typically 37 to 39% (*w*/*w*) glucose and 38–39% (*w*/*w*) fructose, as determined by HPLC [28]. Table 1 shows the aim, the investigated parameters and the setup of the individual experiments. Before each experiment, the liquid honey was homogenized in a 300 kg stirring machine and then divided into the 50 kg stirring machines for each experimental setup. An amount of 25 kg of honey was stirred per machine. There the honey was stirred until the color changed from transparent brown to yellowish white, and the honey’s viscosity just allowed the honey to be filled in glass jars [26]. In order to determine the end of the stirring process, we used a modified version of the “finger test” commonly used by beekeepers: a plastic tongue depressor of 15 cm × 1.8 cm size was dipped 2 cm deep into the honey and pulled 10 cm, which formed a groove. The viscosity was considered suitable when the 2 walls of the groove touched each other after 5 s (Appendix A).

After stirring, the honey was filled in jars and stored for several weeks. We determined the creaminess of the honey on different days using the parameters crystal size, whiteness index and mouthfeel in the following way.

### 2.2. Determination of the Crystal Size by Microscopic Analysis and the Whiteness Index

The crystal size was determined as described previously [32] with three replicates per sample. For the determination of the whiteness index, the honey sample was transferred to a petri dish, and the color was determined via a CIE LAB instrument (Spectro-Colour, Type LMG183, Dr. Lange) according to the manufacturer’s instructions. The whiteness index was calculated by the following formula [19,25,33,34]:
*WI* = 100 – [(100 – *L*
^*^)^2^ + *a*
^*2^ + *b*
^*2^]^0.5^


The determination was repeated three times per honey sample.

### 2.3. Determination of the Mouthfeel by Sensory Analysis

Trained testers investigated the mouthfeel of the honey samples in sensory cabins under standardized conditions. The honey samples were offered coded with random numbers in standardized glasses (30 ml). In stirring tests 1 and 2, the testers assessed the samples by ranking their creaminess. In stirring tests 4 and 5, the testers determined the mouthfeel on a scale between 0 and 100, where 0 was defined as velvety creamy, like hazelnut cream “Nudossi,” and 100 was defined as coarsely crystalline, like crunchy chocolate cream “Ovomaltine Crunchy Cream,” using sensory software RedJade. Beforehand several chocolate creams and hazelnut creams were tested for the definition of scale 0 and scale 100. Due to Corona contact restrictions, sensory testing could not be carried out for stirring test 3.

## 3. Results

### 3.1. Influence of the Stirring Device on Creaminess

We hypothesized that the stirring device might have an influence on the creaminess of the treated honey. We stirred equal quantities of the same honey batch manually or with a stirring spiral in a bucket and with a stirring impeller or a stirring screw in a stirring machine (Appendix A shows the stirring devices). Furthermore, one aliquot of honey was left unstirred as a control. In this experiment, the temperature was kept at 17.5 ± 1.5 °C. We measured the crystal size and the whiteness index at the end of stirring and at several time points in the following 56 days. Three days after the end of stirring, the unstirred honey, as well as the samples stirred with the spiral or the impeller, had significantly larger crystals than the honey stirred manually or with the stirring screw (ANOVA and Student–Newman–Keuls procedure, α = 0.05; Figure 1a), groups with significantly different crystal sizes are marked with “a” and “b”). After 56 days of storage, the crystal sizes became smaller and equal in all experimental setups so that no significant differences could be found (ANOVA, alpha = 0.05).

While the crystal size decreased over time, the whiteness decreased also. At the end of the stirring process, unstirred honey was significantly whiter than the other samples (ANOVA and Student–Newman–Keuls procedure, α = 0.05). Supplementary to this, 56 days after the end of stirring, the unstirred honey and the sample stirred with the stirring impeller were whiter than the other samples (ANOVA and Student–Newman–Keuls, alpha = 0.05; Figure 1b). The decrease in whiteness during honey storage is consistent with the findings of Visquert [34] as well as Radtke and Lichtenberg–Kraag [26]. The mouthfeel was tested after 56 days of storage by 54 sensory testers who ranked the samples in creaminess. The statistical analysis detected no significant differences in the mouthfeel of the honey samples (Friedmann Test, α = 0.05; Figure 1c).

### 3.2. Influence of Pretreatment of Honey on Creaminess

Previous data have shown that crystallization can be best controlled when there are no huge crystals in the honey, and its viscosity is low to facilitate homogenization [14]. Prior to stirring, honey is frequently pretreated by sieving and/or mild heating. To investigate the potential effects of such measures, honey was either used untreated or sieved through a conic pointed sieve with a mesh size of 200 µm with or without having short contact with a heating coil of 55 °C (Appendix A).

Subsequently, the honey was stirred with a stirring spiral at 17.5 ± 1.5 °C two times per day for 15 min. Analysis of the crystal size revealed no significant differences directly after stirring (Figure 2a). However, the crystal size remained relatively constant over time in the batch that had been sieved and heated, while an increase of crystal size was observed in the other variants. Twenty eight days after stirring end, the honey pretreated with sieve and heating coil had significantly smaller crystals than the other samples (ANOVA and Student-Newman-Keuls, alpha = 0.05).

The whiteness index increased while stirring the honey (Figure 2b). Twenty-eight days after stirring ended, there was no significant difference between the pretreatment groups (ANOVA, alpha = 0.05).

In line with crystal analysis, sensory testing showed that pretreatment with a sieve and heating coil led to a significantly creamier mouthfeel than pretreatment only with the sieve or no treatment at all (Friedmann test, alpha = 0.05; Figure 2c).

### 3.3. Influence of the Stirring Temperature and the Addition of Seed Honey on Creaminess

To initiate crystallization, the addition of already crystallized seed honey is frequently recommended in such a way that the fine seed crystals will act as primary crystallization nuclei [14,24,25,26,27,33,35]. In addition, the seed honey may be pretreated to reduce the size of the crystallization nuclei [24,27,36]. Subramanian et al. [13] described that small air bubbles could provoke nucleation and crystallization. In a preliminary experiment, we investigated different ways of preparing the seed honey and its impact on the creaminess of the stirred honey batches.

For the kitchen food processor method (recommended by a local beekeeper), about 2.5 kg of honey rich in glucose of a creamy consistency were stirred for 5 min with a dough hook attachment followed by a whisk attachment at 150 rpm in a kitchen food processor. As a result, tiny air bubbles with a maximum diameter of about 30 µm were created. Alternatively, the seed honey was prepared using a stirring machine equipped with an impeller in such a way that the impeller was not covered with honey, and small air bubbles were stirred into the honey. These seed honeys were added to fresh honey, which was then stirred at 17.5 ± 1.5 °C two times per day for 15 min. At the end of stirring, honey seeded with the kitchen food processor-treated honey had smaller crystals (59 µm) than honey seeded with the honey prepared by stirring with an impeller (101 µm) or honey stirred without seed honey (146 µm). Thus, the preparation of seeding honey with a kitchen processor gave the best results. This method was used for the subsequent experiments. Dyce [24] recommends storing honey at 14 °C after adding the seed honey. Al-Habsi et al. [36] report that 14 °C is the most frequently used crystallization temperature applied in practice and leads to linear growth of the crystal content of the honey. Here we wanted to investigate the consequence if the stirring temperatures deviate from 14 °C for the crystallization of seeded and non-seeded honey. We stirred the honey with a stirring screw at different temperatures, seeded with honey prepared with the kitchen food processor or without seed honey. We found that all chilled samples had huger crystals right at the end of stirring compared to 14 or 28 days later (Figure 3a), where the crystal size decreased. After 28 days, the honey stirred at room temperature showed significantly (ANOVA and Student–Newman–Keuls, alpha = 0.05) larger crystals than the honey stirred at 14 °C or 18 °C (Figure 3a). The addition of seed honey showed no influence on crystal size for honey stirred at 14 °C, whereas honey stirred at 18 °C had significantly smaller crystals when seeded. These results confirm the findings of Costa et al. [25] and Radtke & Lichtenberg–Kraag [26]. The whiteness is at the end of stirring, as well as 28 days later, significantly higher for honey stirred at 14 °C with seed honey compared to the other samples (ANOVA and Student–Newman–Keuls, alpha = 0.05; Figure 3b).

Due to Corona restrictions, sensory testing could not be carried out for this experiment.

### 3.4. Influence of the Stirring Interval on Creaminess

To this end, our results showed that a temperature of 14 °C during stirring is the most critical parameter for improving honey creaminess. Next, we wanted to investigate whether the stirring interval influences the creaminess of honey because, for many hobbyist beekeepers, it is difficult to implement several stirring processes throughout the day. To investigate the stirring process under this condition in more detail, non-seeded honey was stirred at 14 °C for different intervals but with the same total stirring time per day.

At the end of stirring, the crystals were relatively large, and no significant differences (ANOVA, α = 0.05) in the crystal sizes could be detected (Figure 4a). Upon storage, the honey became heavily crystallized, which impeded the separation of intact crystals and thereby determination of the crystal size at later time points. The stirring interval had a very small impact on the whiteness index (Figure 4b). Twenty-three days after stirring ended, the difference in the whiteness index was not significant (ANOVA, alpha = 0.05). Due to Corona contact restrictions, sensory testing could be carried out only with a small sensory panel of five trained testers. The testers determined the mouthfeel on a scale between 0 (velvet creamy) and 100 (coarsely crystalline), as described above. The results show large deviations in the mouthfeel due to the different testers’ assessments (Figure 4c). On none of the days could significant differences in the mouthfeel of the samples be detected (ANOVA, alpha = 0.05). These results indicate that, at a temperature of 14 °C, the stirring interval has no relevant impact on the creaminess of spring blossom honey.

### 3.5. Influence of the Stirring Time on Creaminess

Having established that the stirring interval is of minor importance, we investigated whether the duration of stirring is a critical factor. We stirred the honey samples without seed honey at a temperature of 14 °C.

At the end of stirring, we found a significant influence of the stirring time on the crystal size. Honey stirred daily twice for 15 min had significantly smaller crystals than the other samples (ANOVA and Student–Newman–Keuls, alpha = 0.05; Figure 5a). Thirty days after the end of stirring, the crystal sizes had declined, and the difference between the stirring groups was diminished.

The whiteness index showed little difference between the samples (Figure 5b). At the stirring ended, the sample stirred daily twice for 15 min was the whitest one (ANOVA and Student–Newman–Keuls, α = 0.05). Twenty days later, there was only little impact of the stirring time on the whiteness index (Figure 5b). Due to Corona restrictions, the sensory analysis could be carried out only once with five testers 20 days after the stirring ended. The sensory testers evaluated the mouthfeel of all samples as very creamy. No significant differences could be detected between the kinds of honey with different stirring times (ANOVA, α = 0.05: Figure 5c).

This experiment demonstrated that at a temperature of 14 °C, the stirring time has no significant impact on the creaminess of the spring blossom honey.

## 4. Discussion

It is well established that the crystallization behavior of honey depends on its composition, mainly the glucose to fructose ratio [9,14,27], which varies greatly depending on the foraging flower and from year to year and is difficult to predict. However, it has been shown that also stirring [24,25] and the storage temperature [26] significantly impact the size of the crystals formed. Moreover, the addition of seed crystals is a common practice to reduce crystal size [27]. Stirring of honey is a complex process in which several factors determine the creaminess of the end product: the stirring device, interval and time as well as pretreatment with a sieve and/or heating coil, and the addition of seed honey (Figure 6).

To investigate the impact of these factors in more detail, we performed a series of experiments where we varied the different parameters. Interestingly, the type of stirring device had only a small impact on crystal size, which, moreover, became equal upon storage. Similarly, also pretreatment with a sieve or heating coil affected crystal size, whiteness index and sensory testing only slightly. In sharp contrast, the stirring temperature turned out to be the most important factor, and the smallest crystals were obtained at 14 °C. Interestingly, the stirring time and interval at that temperature had minor effects.

Thus, stirring at the optimal temperature of 14 °C allows other factors to be neglected. Our data also show that in case a temperature of 14 °C during stirring cannot be maintained, seeding with creamy, crystallized honey mixed using a kitchen food processor is recommended. A stirring machine with a screw stirrer, stirring several times per day, as well as pretreatment with a sieve and heating coil, may help to get small crystals.

We were surprised that the crystals became smaller during storage after stirring. We have no physical explanation for this phenomenon. The relation between storage time and crystal size does not seem to be linear but resembles the decay equation. Further research is recommended to investigate this phenomenon in more detail. However, this observation suggests that stirring honey and subsequently keeping it in a cool place for a few weeks before selling might be a helpful practice to obtain a product with the texture preferred by the consumers.

## Figures and Tables

**Figure 1 foods-12-00048-f001:**
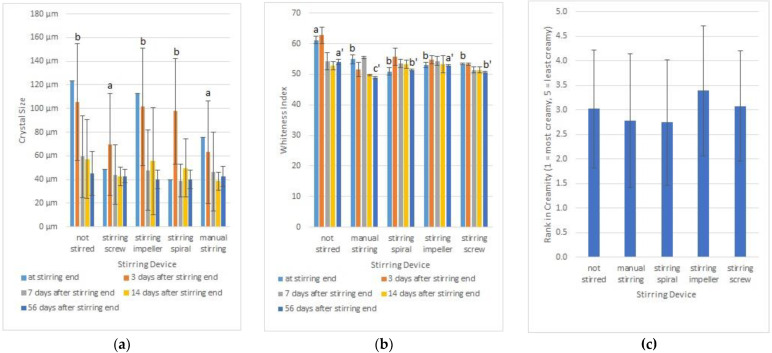
Influence of stirring device on the creaminess of honey. (a, b, a’, b’, c’ denote groups of significantly different samples.) (**a**) Mean and standard deviation of the size of the 10 largest crystals on different days. (**b**) Mean and standard deviation of whiteness index of the honey on different days. (**c**) Mouthfeel of the honey, tested by 54 trained sensory testers by ranking test after 56 days storage (average rank and standard deviation).

**Figure 2 foods-12-00048-f002:**
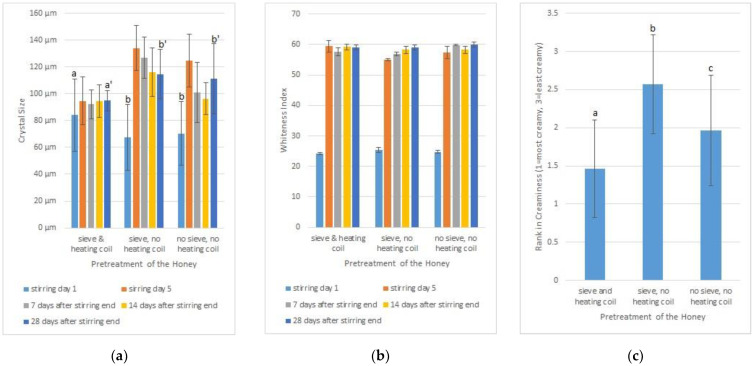
Influence of the honey pretreatment on its creaminess. (a, b, c, a’, b’ denote groups of significantly different samples.) (**a**) Mean and standard deviation of the size of the 10 largest crystals on different days. (**b**) Mean and standard deviation of whiteness index of the honey on different days. (**c**) Mouthfeel of the honey, tested by 41 trained sensory testers by ranking test (average rank and standard deviation).

**Figure 3 foods-12-00048-f003:**
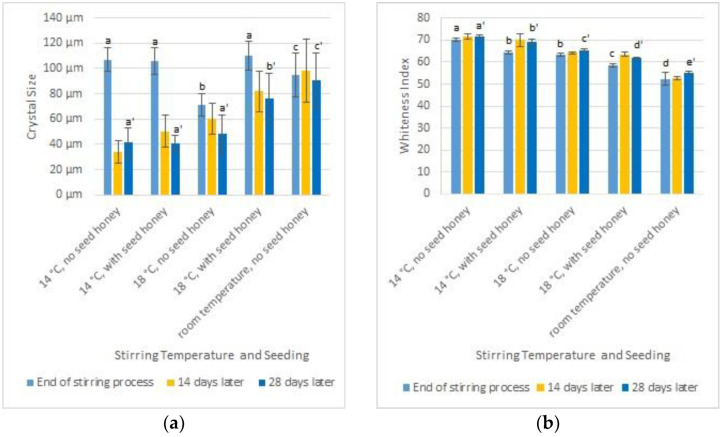
Influence of the stirring temperature and the addition of seed honey on the creaminess of honey. (a, b, c, d, a’, b’, c’, d’, e’ denote groups of significantly different samples.) (**a**) Mean and standard deviation of the size of the 10 largest crystals on different days. (**b**) Mean and standard deviation of whiteness index of the honey on different days.

**Figure 4 foods-12-00048-f004:**
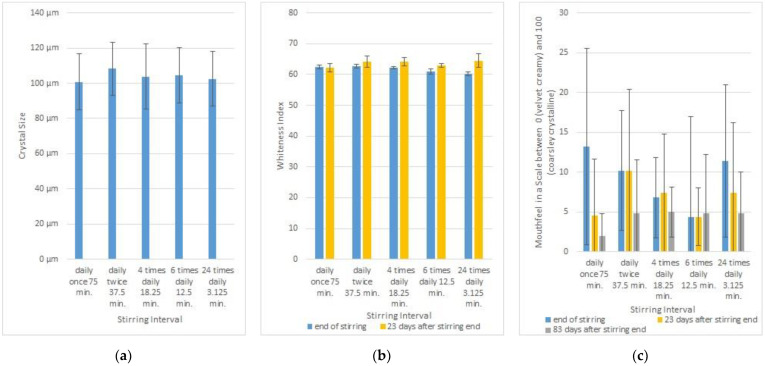
Influence of the stirring interval on the creaminess of honey. (**a**) Mean and standard deviation of the size of the 10 largest crystals on different days. (**b**) Mean and standard deviation of whiteness index of the honey on different days. (**c**) Mouthfeel of the honey, tested by 5 trained sensory testers on a scale between 0 (velvet creamy) and 100 (coarsely crystalline; average scale and standard deviation).

**Figure 5 foods-12-00048-f005:**
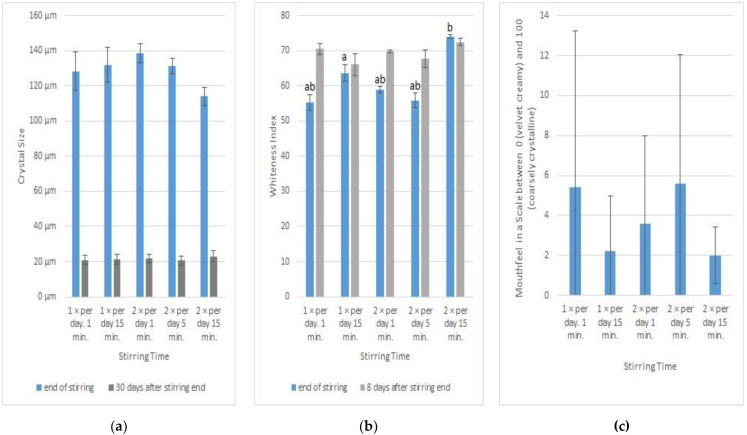
Influence of the stirring time on the creaminess of honey. (a, b denote groups of significantly different samples.) (**a**) Mean and standard deviation of the size of the 10 largest crystals on different days. (**b**) Mean and standard deviation of whiteness index of the honey on different days. (**c**) Mouthfeel of the honey, tested 20 days after stirring end by five trained sensory testers on a scale between 0 (velvet creamy) and 100 (coarsely crystalline; average scale and standard deviation).

**Figure 6 foods-12-00048-f006:**
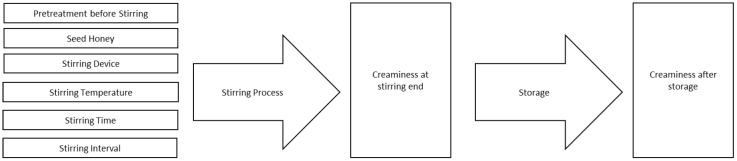
Interaction of the stirring process parameters and storage to the creaminess of honey rich in glucose.

**Table 1 foods-12-00048-t001:** Aim, variants and parameters of the stirring tests.

No.	Aim and Variants	Pretreatment before Stirring	Seeding	Stirring Device	Stirring Time	Stirring Interval	Stirring Temperature	Storing Temperature after Stirring
1	Influence of the stirring device on creaminess [29] unstirred stirring machine with stirring screw stirring machine with impeller stirring machine with stirring spiral manual stirring	Sieve and heating coil	7% seed honey	See column aim and variants	15 min	3× per day	16–19 °C	14 °C
2	Influence of honey pretreatment on creaminess [29] sieve and heating coil sieve, no heating coil no sieve, no heating coil	See column aim and variants	7% seed honey	Stirring spiral	15 min	2× per day	14–16 °C	14 °C
3	Influence of seeding and stirring temperature on creaminess [30] stirred at 14 °C with seed honey stirred at 18 °C with seed honey stirred at 14 °C, without seed honey stirred at 18 °C, without seed honey stirred at room temperature, without seed honey	Sieve and heating coil	If seeded: seed honey prepared with a kitchen food processor	Stirring screw	15 min	3× per day	See column aim and variants	Same temperature as the stirring temperature
4	Influence of the stirring interval on creaminess [31] daily once 75 min. daily twice 37.5 min. 4 times daily 18.25 min. 6 times daily 12.5 min. 24 times daily 3.125 min.	Sieve and heating coil	None	Stirring screw	See column aim and variants	See column aim and variants	14 °C	14 °C
5	Influence of pretreatment and total stirring time on creaminess [31] stirring 1× per day 1 min. stirring 1× per day 15 min. stirring 2× per day 1 min. stirring 2× per day 5 min. stirring 2× per day 15 min.	Sieve and heating coil	None	Stirring screw	See column aim and variants	See column aim and variants	14 °C	14 °C

## Data Availability

The data will be available on request.

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
