# Peer review of "Influence of Stirring Parameters on Creaminess of Spring Blossom Honey Measured by Crystal Size, Whiteness Index and Mouthfeel"

_foods, 2022, doi:10.3390/foods12010048_

Round 1

Reviewer 1 Report

Manuscipt foods-2111725 is a short communication article that describes how the sensory characteristics of honey (colour, mouthfeel, texture), and especially its texture, is affected by stirring and storage time. The authors provided data for glucose contents as its crystals affect greatly honey texture during storage or processing. The manuscript is relevant to the aims and scope of Foods journal. The concept of the study is novel and data are provided for Spring blossom honey harvested in Germany.

The design of the study is adequate and the conclusions are supported by the data shown. However, there are some points that authors should revise in order to improve further the general flow of the article and data presentation.

I have enclosed within the attached pdf my comments.

Based on these comments, I suggest a minor revision prior.

Author Response

Reviewer 1

We want to thank reviewer 1 for the helpful comments. We have given each point careful consideration as described below. The original text of reviewer 1 is in black while our answers are in blue. Changes in the manuscript are highlighted yellow.

Manuscipt foods-2111725 is a short communication article that describes how the sensory characteristics of honey (colour, mouthfeel, texture), and especially its texture, is affected by stirring and storage time. The authors provided data for glucose contents as its crystals affect greatly honey texture during storage or processing. The manuscript is relevant to the aims and scope of Foods journal. The concept of the study is novel and data are provided for Spring blossom honey harvested in Germany.

The design of the study is adequate and the conclusions are supported by the data shown.

Answer 1

Thank you for reviewing our paper and for the positive evaluation.

However, there are some points that authors should revise in order to improve further the general flow of the article and data presentation.

I have enclosed within the attached pdf my comments.

Based on these comments, I suggest a minor revision prior.

Answer 2

Thank you for the helpful remarks in the manuscript. We have corrected all formal mistakes. In addition, we have improved the discussion and included the information typically given in a conclusion in the discussion part to keep the manuscript, which is a communication, short.

Reviewer 2 Report

I recommend it needs some minor clarifications are done.

 1. All image formats need to be uniform, significance analysis should be completely supplemented, coordinate ordinate should be optimized, is it reasonable to have such a wide difference in sensory evaluation?

2. Please modify and unify the reference format, such as”1” and “19”, etc.

3. The discussion section needs to be reorganized and improved.

Author Response

Reviewer 2

We want to thank reviewer 2 for the helpful comments. We have given each point careful consideration as described below. The original text of reviewer 2 is in black while our answers are in blue. Changes in the manuscript are highlighted yellow.

I recommend it needs some minor clarifications are done.

  1. All image formats need to be uniform, significance analysis should be completely supplemented, coordinate ordinate should be optimized, is it reasonable to have such a wide difference in sensory evaluation?

Answer 1

We agree that the format of the figures should be uniform. We have improved the style of the figures and the statistical analysis. We have included table S1 with the significance analysis of all findings.

  1. Please modify and unify the reference format, such as”1” and “19”, etc.

Answer 2

Thank you for this comment. We have modified the references as suggested.

  1. The discussion section needs to be reorganized and improved.

Answer 3

Thank you for this suggestion. We have reorganised and improve the discussion.

Reviewer 3 Report

The manuscript mainly discusses the influence of Stirring Parameters on Creaminess of Spring Blossom Honey Measured by Crystal Size. In addition, the authors studied Whiteness Index and Mouthfeel. Though this is a good study, there are other factory that effect the creaminess of honey.

1.   Include values in the abstract.

2.   Why did the authors only focus on creaminess? As other parameters could effect the quality of honey.

3.   Emphasize the novelty of the study.

4.   Why did the authors did not look at viscosity?

5.   Did the authors measure the diameter of the crystals?

6.   How would the stirring affect the nutritional value of honey?

Author Response

Reviewer 3

We want to thank reviewer 3 for the helpful comments. We have given each point careful consideration as described below. The original text of reviewer 3 is in black while our answers are in blue. Changes in the manuscript are highlighted yellow.

The manuscript mainly discusses the influence of Stirring Parameters on Creaminess of Spring Blossom Honey Measured by Crystal Size. In addition, the authors studied Whiteness Index and Mouthfeel. Though this is a good study, there are other factory that effect the creaminess of honey.

  1. Include values in the abstract.

Answer 1

We have included important experimental data and values.

  1. Why did the authors only focus on creaminess? As other parameters could effect the quality of honey.

Answer 2

We agree that many factors impact on honey quality. For instance, besides creaminess the glucose to fructose ration, the water, salt, and 5-hydroxymethylfurfural (HMF) content and the activity of a number of enzymes including diastase and invertase are important. However, stirring impacts only on sugar crystal growth while it does not alter the chemical composition. Thus, stirring has a major impact on the crystal size while it little to no effect the other parameters. Moreover, many studies have focused on other quality parameters like the HMF content while only a few studies have investigated crystallisation. Thus, we decided to focus on this aspect.

  1. Emphasize the novelty of the study.

Answer 3

As mentioned above, only a few studies have investigated crystallisation of honey so far and most of them used laboratory conditions, i.e. small sample sizes. In contrast, we used large quantities (25 kg for each assay), which fits well to the batch size used in beekeeper´s practice. We have adapted the introduction to emphasize hat.

  1. Why did the authors did not look at viscosity?

Answer 4

We agree that viscosity is an important parameter. However, crystallised honey is a biphasic system consisting of a highly viscous liquid and more or less huge crystals. The viscosity of the liquid is mainly determined by the sugar content of the liquid phase rather than the size of the crystals suspended in it. In contrast, the mouthfeel (coarse or creamy) is mainly determined by the crystal size. Thus, we decided to focus on that parameter.

  1. Did the authors measure the diameter of the crystals?

Answer 5

The length of the crystals has the greatest impact on the mouthfeel and thus we decided to measure this parameter. We did not assess the crystal diameter.

  1. How would the stirring affect the nutritional value of honey?

Answer 6

We do not think that stirring has an impact on the nutritional value. As we show in our manuscript, stirring impacts on the crystal size. In contrast the chemical composition of honey is not altered. Upon consumption honey crystals melts 
